# Role of Food Industry in Promoting Healthy and Sustainable Diets

**DOI:** 10.3390/nu13082740

**Published:** 2021-08-10

**Authors:** Kevin B. Miller, James O. Eckberg, Eric A. Decker, Christopher P. F. Marinangeli

**Affiliations:** 1General Mills, Bell Institute of Health and Nutrition, Global Scientific & Regulatory Affairs, Minneapolis, MN 55427, USA; 2General Mills, Nutrition Technology and Science, Minneapolis, MN 55427, USA; jim.eckberg@genmills.com; 3Department of Food Science, University of Massachusetts, Amherst, MA 01003, USA; edecker@foodsci.umass.edu; 4Pulse Canada, 920-220 Portage Ave, Winnipeg, MB R3C 0A5, Canada; cmarinangeli@pulsecanada.com

**Keywords:** sustainable, diets, packaged, food, consumer, affordable

## Abstract

Sustainable food systems are often defined by greenhouse gases, land use, effects on biodiversity, and water use. However, this approach does not recognize the reason food is produced—the provision of nutrients. Recently, the relationship between diets and sustainability has been recognized. Most accepted models of ‘sustainable diets’ focus on four domains: public health, the environment, food affordability, and cultural relevance. Aligned with the FAO’s perspective, truly sustainable diets comprise foods that are affordable, nutritious, developed with ingredients produced in an environmentally friendly manner, and consumer preferred. Identifying solutions to address all four domains simultaneously remains a challenge. Furthermore, the recent pandemic exposed the fragility of the food supply when food accessibility and affordability became primary concerns. There have been increasing calls for more nutrient-dense and sustainable foods, but scant recognition of the consumer’s role in adopting and integrating these foods into their diet. Dietary recommendations promoting sustainable themes often overlook how and why people eat what they do. Taste, cost, and health motivate consumer food purchase and the food system must address those considerations. Sustainable foods are perceived to be expensive, thus marginalizing acceptance by the people, which is needed for broad adoption into diets for impactful change. Transformational change is needed in food systems and supply chains to address the complex issues related to sustainability, taste, and cost. An emerging movement called regenerative agriculture (a holistic, nature-based approach to farming) provides a pathway to delivering sustainable foods at an affordable cost to consumers. A broad coalition among academia, government, and the food industry can help to ensure that the food supply concurrently prioritizes sustainability and nutrient density in the framework of consumer-preferred foods. The coalition can also help to ensure sustainable diets are broadly adopted by consumers. This commentary will focus on the challenges and opportunities for the food industry and partners to deliver a sustainable supply of nutrient-dense foods while meeting consumer expectations.

## 1. Introduction

While sustainable diets have become an increasingly frequent topic for debate, a global definition and the parameters that encompass a sustainable diet remain unclear. The FAO has recognized that the lack of a common definition of sustainable diets exacerbates the complexity of the topic because their perspective of sustainable diets extends beyond nutrition and environmental impacts to include both economic and socio-cultural dimensions [1]. Although various dietary guidelines aim to incorporate aspects of sustainability into healthy dietary patterns, the domains of sustainability differ between regions. For example, Sweden’s National Food Agency (Livsmedelsverketthat) integrated health and environment into their nutritional advice [2]. Brazil’s dietary guidelines are built upon a set of principles that acknowledge the interdependence between healthy diets and the social and environmental sustainability of the food system [3]. One commonly cited illustration (Johnston et al.) captured the complexity of sustainable diets, where aspects of nutrition and health, culture, pleasure, equity, well-being and health, environment, and biodiversity protection are equally weighted (Figure 1) [4]. From Figure 1, it is understandable that the complexity of sustainable diets is a barrier to the development of clear recommendations for use in dietary guidelines. One underlying theme of existing dietary guidance is the notion that sustainability is best exemplified by fresh, local, and unprocessed foods. There is merit in recommending that the public increase their consumption of such foods, including fruits, vegetables, legumes, and whole grains. However, recommendations have also been presented to avoid processed food which may have the unintended consequence of limiting the accessibility of nutrient-dense foods to populations with limited buying power, access to fresh foods (e.g., food deserts), and cooking time and skills. Unfortunately, this recommendation has caused ‘processed food’ to have a broadly negative connotation despite the fact that food processing can also improve the nutrition attributes of foods and increase the incorporation of nutrient-dense products into daily diets. Processed foods include products across a wide spectrum and many include food groups to be encouraged, including fruits, vegetables, legumes, whole grain, nuts and seeds, and low-fat dairy. Provided foods can be sustainably produced and nutrient dense, processing per se should not negatively impact whether or not the food is recommended because this reduces consumer options.

Packaged foods can be widely accessible, affordable, and consumed by the majority of the American population [5]. The critical role that the food industry can play in bringing sustainably sourced, nutrient-dense foods to the majority of consumers cannot be over-emphasized, especially when considering the influence that the food industry can have on the entire supply chain. The food industry can provide drivers for the production of novel commodity crops, raw materials, and ingredients using more sustainable practices. There is merit to the recommendations of fresh and local when practical (seasonality), accessible (geographic location), and affordable to the average consumer. However, the ability to efficiently distribute inexpensive, shelf-stable, nutrient-dense foods across the country (or globe) highlights the opportunity for packaged foods to play an important role and contribute toward higher diet quality.

Packaged foods can help provide access to healthy, that is, nutrient-dense, foods regardless of economic status. The wide accessibility of packaged foods also represents an opportunity to deliver large amounts of critical nutrients. There are numerous success stories of the importance of essential vitamin and mineral fortification of packaged foods that have widespread consumption (e.g., salt, milk, flour) which leads to improved health outcomes across a broad spectrum of the population.

An example of this successful strategy is the fortification of grain products with folate that has resulted in a large reduction in neural tube defects in infants (FDA and CDC). Packaged foods can also deliver large quantities of nutrients like fiber. Whole grains are an example of a packaged food whose processing increases their nutritional quality. Whole grain consumption was recommended as the predominant source of daily calories in the EAT-Lancet’s Planetary Health Diet [6] and is included in most dietary guidance around the world (including Australia, Canada, Denmark, France, Germany, Mexico, Singapore, Spain, Sweden, the UK, and the United States). Seeds are largely non-digestible, so their processing is essential to unlocking their nutrition potential. Whole grains are almost universally processed, and this processing can make the grains more palatable, stable, nutritious, easy to prepare, safe, and less expensive. Therefore, it is important to re-evaluate recommendations that processed food are unhealthy. Educating and encouraging consumers to identify foods from all sources that are nutrient-dense, affordable, and responsibly/sustainably produced should be a priority in public health initiatives and dietary guidance regardless of whether they are fresh, frozen, canned, or otherwise packaged.

## 2. Discussion

### 2.1. Defining Sustainability for Consumers

The food industry communicates with consumers through advertising and on-package information. In some countries, packaging includes data or schemes representing the product’s calculated impact on the environment. Communication of a food’s sustainability is often limited to a description of greenhouse gas emissions, or the common vernacular ‘carbon footprint’, as found in the UK [7], France [8], and Germany [9]. Whether or not consumers are influenced by the communications identifying a product’s environmental ‘footprint’ even when the foods are not equally nutritious is unknown. Use of these sustainability schemes represents some complications for the consumer and within the food industry. Nutrient-dense foods often have a larger environmental footprint. How can the consumer understand whether or not two foods are equally nutritious? Such insight is important for consumers to evaluate potential tradeoffs in nutrient density and sustainability among food options. For example, a published study reported that dairy milk had greater environmental impact on greenhouse gases (GHGs), water, and land use than dairy alternative beverages from almond, oat, and rice [10]. These data are often cited and potentially misunderstood as evidence why consumers should choose plant-based alternative products. Over-simplified sustainability communications sometime ignore the differences in protein content, protein quality, calcium, and other shortfall nutrients provided by one option and not others. Whether or not the consumer understands the nuances of the nutrient quality of these products is not equivalent and the influence on their purchasing decision is unknown. Nutrient density of a food can serve as a common denominator when making decisions among foods and beverages [11]. Unless nutrition is considered as the common denominator when evaluating these schemes, it is impossible to fairly assess both environmental and health impact. As illustrated in Figure 1, ‘sustainable diets’ is an encompassing term. Truly sustainable diets are those that prioritize nutrient-dense foods produced using sustainably sourced ingredients and not a diet solely comprising foods with the lowest carbon footprints.

As consumers become increasingly conscious of sustainability in their shopping, it will be necessary to reinforce the concept of nutrition to avoid choices being made without understanding the full value of the food. The most sustainably grown crops/foods are not actually sustainable if they do not make significant and positive contributions towards health and nutrition. Consumer surveys indicate that most of the public have their own concept of what comprises a sustainable food. A 2020 survey of American consumers reported that approximately 40% identified sustainable foods as organic while more than 40% indicated sustainable foods were locally grown [12]. Seventy-six percent of those surveyed indicated their personal health was the primary reason for purchasing organic. Common reasons cited included concern about the potential ingestion of pesticides, herbicides, and chemical fertilizers, as well as animal welfare. The impact of pesticides and herbicides on the environment did not appear to be a key motivation of purchase. Another survey reported that consumers defined sustainable foods as organic, natural, and non-GMO [13]. These points help illustrate and reinforce that some consumers’ perspectives of sustainable foods are aligned with their personal needs and do not actually always align with sustainability. Communicating with consumers on the topic of sustainable diets will be challenging without a common definition among the public, industry, policy makers, and non-government organizations. Until then, the food industry will have to ensure that the sustainability messages shared are appropriate, accurate, and consumer relevant and do not perpetuate misperceptions.

Although the challenge of balancing nutrition and sustainability has been described [14], the inclusion of nutrition into the conversation of sustainability is still relatively recent and spurred by the need to better define sustainable diets [15]. The published report from the EAT-Lancet Commission on healthy diets from sustainable food systems [16] could be credited with bringing the spotlight onto nutrition in the context of sustainability. The report included a recommended diet based as much on nutrition and health outcomes as environmental impact. The results were, not surprisingly, consistent with the majority of national dietary guidance encouraging a diverse diet that emphasizes fruits, vegetables, whole grains, and nuts. How these recommendations to consume a nutrient-dense, sustainably sourced diet will be effectively communicated to consumers and translated into action will require the collaboration of all sectors, public and private.

### 2.2. The Consumer and Shopping with a Sustainable Mindset

Dietary Guidelines for Americans (2020–2025) do not currently identify sustainability of the diet as a goal [17]. In fact, the word ‘sustainable’ is absent from the report. However, many of the recommendations in the DGA share significant overlap with EAT’s Planetary Health Diet. The European Union’s Farm-2-Fork initiative includes strategies to promote a ‘fair, healthy, and environmentally friendly food system’ [18]. The typical consumer in Europe appears to be more aware and invested in selecting foods based on their perception of the food’s sustainability. The Environmental Performance Index lists the top 10 countries as European, and the United States was 24th [19]. More consumer insights are necessary to understand whether consumers limit their comparisons of sustainable foods within a category or whether consumers substitute a nutrient-dense food for an entirely different non-analogous food based on sustainability. One specific consumer hurdle to shopping with a sustainability mindset that limits broad adoption is economics. When prices are higher (perceived or real) for a sustainably sourced food, then consumers are less likely to embrace that food. To elicit an effective change in sustainable dietary practices, there needs to be broad adoption across the country, region, or even globally. Other than improving identification and familiarity of which foods are sustainably produced, the food industry needs a better understanding of the barriers, perceived or real, to consumers adopting more sustainable diets. It is understood that the cost of foods is the central barrier to selection if the food is acceptable for taste. There is a need to advance sustainability in ways that bolster profitability along the value chain to deliver sustainable and affordable food options to consumers.

### 2.3. Consumer Perception of Affordability of Sustainable Foods

As summarized by the FAO, sustainable diets must not only be nutrient dense and produced in a way that is respectful of the environment, but accessible and affordable to all consumers. Unfortunately, foods promoting a strong story of sustainable production may not have a wide audience because of the assumption that sustainably produced foods cost more. For example, consumer perception can be based on the belief that sustainably produced foods are ‘organic’ and the consumer’s experience with organic foods is that they cost more. Organic foods often do cost more as a result of higher labor costs (e.g., manually weeding fields), sorting, certification process costs, and even supply versus demand. However, over the past few decades (American) consumers have grown accustomed to getting more for less. In 2019, Americans spent an average of 9.5% of their disposable personal incomes on food—divided between food at home (4.9%) and food away from home (4.6%) [20]. Between 1960 and 1998, the average share of disposable income spent on food by Americans, on average, fell from 17.0 to 10.1%, driven by a declining share of income spent on food at home [20]. The USDA’s Economic Research Service (ERS) also reported that as household incomes rise, more money is spent on food, but represents an even smaller share of their total budget. In 2019, households in the lowest income quintile spent an average of USD 4400 on food, which represented 36% of their income. In contrast, households in the highest income quintile spent an average of USD 13,987 on food, but it represented only 8% of their income. Quite simply, consumers are becoming increasingly accustomed to getting more for less which has been a major driver in the changes to the food supply. In a recent survey of our local food retailers, foods produced to replicate their traditional counterparts, such as plant-based ‘meats’ and ‘milk’, can cost 35–95% more compared to the animal-based products. Therefore, the food industry faces both opportunities and challenges in delivering products that are sustainably produced while also meeting consumer needs for taste, affordability, and nutrient density. Addressing issues of costs in sustainable foods may be the single most effective way to drive adoption of sustainable food options for consumers. This will help consumers see how the foods they currently enjoy can be sustainably produced and allow them to participate by removing the perceived barriers to entry into sustainable food shopping. The food processing industry is uniquely positioned to help drive changes in the food supply to increase sustainability while maintaining accessibility, affordability, and acceptability.

Unfortunately, promoting a healthy, sustainable diet while making it affordable for everyone has been criticized as two incompatible concepts [21]. Countries with less wealth face even greater financial challenges to meeting healthy, sustainable dietary guidance. It was reported that the cost of the sustainable EAT-Lancet diet for rural India ranged between USD 3.00 and USD 5.00 per person per day in contrast to current dietary intake at a value of USD 1.00 per person per day [22]. The authors also noted challenges to achieving recommended diets are exacerbated by seasonal variations and volatility in food prices [22]. The FAO stated that where countries have national food-based dietary guidelines, there is often a lack of policy coherence on how to ensure the affordability of those diets recommended for nutrition and health. Furthermore, they found no definition of a healthy diet that would be globally affordable; all definitions resulted in similar conclusions.

The FAO report [23] identified three takeaways in their analysis of meeting healthy, sustainable diets for all:Healthy diets are unaffordable for many people. The high cost of nutritious and sustainable foods in places where low-income people live is a major obstacle to the achievement of global development goals.Unaffordability of healthy diets is concentrated in Africa and Southern Asia. While these are known to be hot spots for malnutrition, insufficient attention has been paid to diet quality as a cause of malnutrition in all its forms.Supporting nutrient-adequate and healthy diets requires a combination of higher incomes and lower prices, particularly of diverse nutritious items, making a variety of healthier foods more widely available at lower cost.

Although the FAO may not have been identifying typical American or European consumers in their call for affordable, healthy, and sustainable diets, these populations also need access. In addition, there are low-income populations even in more highly developed countries that also have challenges in accessing healthy and sustainable foods. Therefore, the food industry has the opportunity to help provide all consumers with foods that are nutrient-dense, sustainably produced, and affordable. The food industry is in a unique position to help solve this challenge because large-scale food production is able to produce foods at lower cost. This is because manufacturers source raw materials in bulk and there are economic advantages of buying at scale. In addition, food manufacturers utilize specialized equipment that is more energy efficient and produces higher yields. If waste is produced, then it is at a large scale such that waste can be used to produce additional products that can be sold as retail products or animal feed.

### 2.4. Willingness to Pay

In higher socioeconomic level nations, there is still a hurdle to broad consumer adoption of sustainable foods, primarily the consumer’s willingness to spend more while not necessarily perceiving that they are getting more. Organic foods may be an example where many consumers are unwilling to pay the premium because they do not prioritize the value proposition. As the food industry depends upon and is reactive to consumer needs, understanding the consumer’s thresholds will predict what and how the industry will evolve. The food industry must seek more sustainable solutions that are cost neutral to the typical consumer. A meta-analysis on the willingness to pay for foods claiming to be more sustainable reported that consumers in Europe and Asia were the most willing to pay extra for foods identified as ‘sustainable’ (31.9 and 31.8%, respectively) as compared to North America (25.5%) or those living in Oceania (17.2%) [24]. However, the authors noted that the unexpectedly high willingness to pay reported in part of Asia may have been the result of perceptions of improved food safety (low contamination) and not environmental impact concern.

Consumers are not a homologous group, but represent a broad spectrum of needs and motivations. This can often be seen in a philosophy where consumers are either motivated to address their personal and/or family needs as compared to consumers focused on the impact on and benefit to others, including the environment. Consumers prioritizing their personal or family needs may be more constrained by a lower disposable income and therefore seek to maximize the amount of food they can purchase with limited resources. Consumers prioritizing social impact of their food choice may be more represented by individuals with greater disposable income. However, purchasing foods viewed as more sustainable may also include those with lower incomes and a higher willingness to spend on those foods. Regardless, the definition of sustainable foods is blurred by the consumer view that sustainably grown foods are likely more natural and therefore may have fewer contaminants. Some consumers indicate they are happy to spend more money on environmentally friendly products, but this is not well supported by purchase statistics when examined quantitatively. Accurately understanding the consumer’s purchase motivators is necessary before developing communications on a product’s sustainability. In the US, a survey conducted annually over the past decade queried consumers about their motivators for food purchases. According to the latest International Food Information Council (IFIC) consumer survey, over a period of ten years, consumers consistently reported that the food’s taste was the number one factor of purchase, followed by cost, healthfulness, convenience, and, lastly, environmental sustainability [25]. The food industry, a business, aligns its priorities with the consumer’s needs. Based on these data, the food industry can and should continue to invest in a more sustainable supply chain (e.g., grow crops using regenerative principles), but communicating with the *average* American consumer about reduced environmental impact may not have broad appeal. Focusing on the food’s taste and cost are likely to remain the key consumer-facing messages to the overall population with a gradual introduction of sustainability to introduce the concept to those not previously engaged. By demonstrating how foods the average consumer currently enjoys can be sustainable, the food industry can help reach the critical mass necessary to achieve a more sustainable food supply.

## 3. The Role of the Food Industry in Enhancing Sustainability of Diets

Sustainability initiatives are being created and communicated by nearly all sectors and members of the food industry. These initiatives are both internally motivated, but also being developed in response to external pressures. Internally, the food industry traditionally has sought to increase sustainability because this can help lower costs. For example, increasing the energy efficiency of processing operation, decreasing water usage, and reducing the amount of packaging material both increase sustainability and decrease production costs. The external environment includes consumer expectations, but US consumers indicate sustainability is not a primary motivator of food purchase—despite the growing number of consumers who indicate their interest [25]. Other external forces include investors who expect publicly traded companies to demonstrate they have a clear plan for continued growth and success despite pressures that may threaten supply chain stability or negatively impact the company’s reputation. Retailers have significant influence over the food industry as well as their suppliers. Controlling food loss and waste are recognized as significant challenges to sustainability of the food supply. As a result, ReFed, a collaboration of businesses, nonprofits, and the government, was formed to help reduce food waste in the United States. Their report identified as much as 40% of food waste occurring at retail and 43% with the consumer [26]. In contrast, the manufacturing sector was responsible for 2% of food waste (Figure 2). As mentioned earlier, this is due to the ability of the food industry to maximize yield and find additional uses for waste streams. These data help to illustrate the efficiency of the food manufacturing sector and identify opportunities to address food waste, especially within the retail and consumer levels. Advances in packaging technologies and processing are expected to present solutions to increase shelf-life, making foods more likely to be purchased and consumed, and could produce less waste at the retail and consumer levels. Increasing shelf-life also helps all socioeconomic levels as decreasing waste helps with accessibility (less shopping time) and affordability (less food not consumed, thrown away, and repurchased). As a result, there are some distinct advantages associated with the processing and packaging of foods contributing to environmental impact and the nutrition status of the consumer.

A group of key global retailers and manufacturers convened Champions 12.3 and created the ‘10 × 20 × 30’ initiative with a goal of reducing food waste by one half [27]. The 10 × 20 × 30 represents more than 10 key retailers working with at least 20 food manufacturers to halve food waste by 2030. As demonstrated by numerous industry initiatives to address food waste and emissions associated with food production, sustainability is becoming table stakes where actions are expected to yield clear results. Most often, these commitments are published in each organization’s respective annual responsibility reports with progress updates [28,29]. Examining what goals are accomplished and how and their impact on improving health and sustainability is key.

## 4. Addressing Sustainability and Cost with Sustainable Production

Addressing the interconnected challenges of sustainability and cost will require us to rethink our food systems. An emerging farmer-led movement called regenerative agriculture offers promising potential to address sustainability concerns while potentially not increasing, or even decreasing, production costs. Regenerative agriculture is a set of nature-based principles (e.g., crop diversification, livestock integration) used to restore the farm ecosystem and economics [30]. Regenerative principles can improve the profitability of farms [31]. By adding value to farming operations, regenerative agriculture obviates additional costs being passed along the supply chain to consumers. Further, farmers adopting regenerative principles have done so at a scale (i.e., number of acres) similar to conventional production systems. The scalability and farm profitability offer potential for positive impacts on food supply chains, translating into sustainable, affordable food.

Brazil’s dietary guidance acknowledged that the impact of our food supply production must be considered in creating healthy dietary patterns [32]. Despite the fact that only a handful of national dietary guidelines mention sustainability, the food industry has begun to embrace the concept. Several major food manufacturers and ingredient suppliers have announced investment in sustainable agricultural initiatives, including regenerative agriculture [28,33,34,35]. The scale of the food industry, especially the largest companies, can help to positively support the adoption of regenerative principles to deliver sustainably grown foods to consumers. Regenerative agriculture provides fundamental benefits to food companies. Restoring the farm ecosystem (soil, water, biodiversity) potentially bolsters the resiliency of ingredient supply chains. For example, adoption of regenerative farming principles by individual farmers has been shown to slow and even reverse the degradation of the soil where crops are grown [36]. Healthy soils support crop production, especially during times of extreme drought or flooding, both of which are expected to increase under climate change. These initiatives are not embraced by industry simply to generate positive news, but are vital to the organizations to ensure stable farms that result in a secure supply of ingredients. Farmers are leading the adoption of regenerative agriculture within the agriculture community. The food industry has an important role in supporting this farmer-led initiative. By supporting farmer-led education and assessment of ecosystem and economic benefits, the food industry can further accelerate the adoption of regenerative agriculture to drive positive change in food systems.

## 5. Conclusions

In summary, solutions to deliver healthy, sustainably sourced foods must occur within the context of the consumer who values taste, cost, and convenience above all other food attributes. Sustainably sourced, nutrient-dense packaged foods may help to engage consumers who do not prioritize, feel they cannot afford, or simply lack access to ‘fresh’ and ‘locally sourced’ foods that are traditionally perceived as being sustainable. Transitioning everyone in the population to a more sustainable food supply will be accomplished in a shorter time by recognizing that the food industry has the scale and influence to make significant changes to what and how we eat, especially regarding consumers who may not have the resources or interest to pursue ‘sustainable’ foods independently. Sustainable, nutrient-dense foods can come from all sources, including processed foods. Informing the typical consumer and helping them to identify and select widely available foods that are affordable and convenient, but still nutrient dense and sustainably sourced, will likely be more effective than recommendations to avoid foods they currently consume and enjoy. Development of tools and metrics that assess the environmental impact and nutrient density of foods will be critical to appropriately shaping the food supply of the future. The food industry has a responsibility for delivering consumer-preferred foods while encouraging their suppliers (i.e., ingredients, packaging) to adopt processes and practices that reduce their respective environmental footprints while remaining competitive with their peers and appealing to investors. A broad coalition of governments, academics, and industry can collaborate to overcome hurdles of consumer perception and develop education and communications that will help consumers identify and adopt sustainable, nutrient-dense foods in their diets

## Figures and Tables

**Figure 1 nutrients-13-02740-f001:**
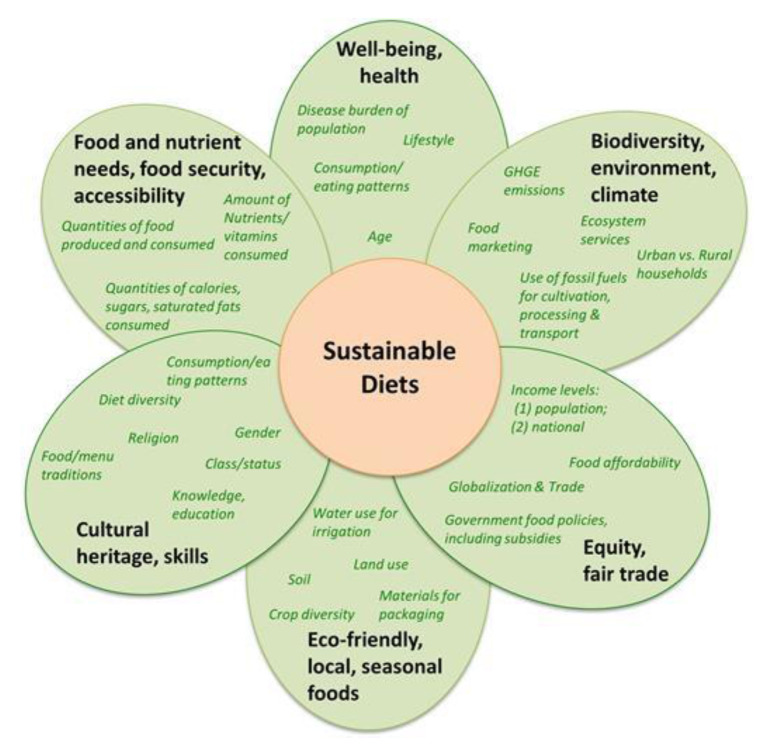
Schematic representation of the key components of a sustainable diet (Johnson et al. 2014).

**Figure 2 nutrients-13-02740-f002:**
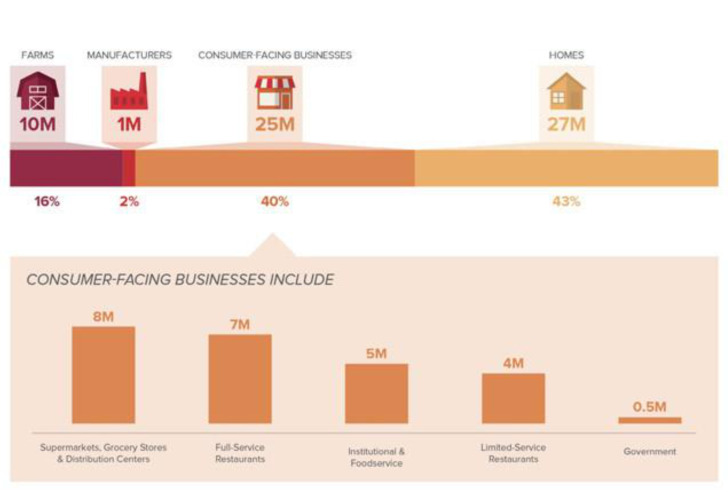
Origin and quantity of food waste in the United States, identifying the opportunities for food manufacturers to help retailers and consumers reduce their waste (ReFed).

## Data Availability

Not applicable.

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
