# Peer review of "Role of Food Industry in Promoting Healthy and Sustainable Diets"

_nutrients, 2021, doi:10.3390/nu13082740_

Round 1

Reviewer 1 Report

This is a well written and well sourced document on a particularly important topic.  It is clear from the commentary that there are a complex set of issues that need to be considered in defining, promoting and achieving a sustainable, healthy diets.  Simple solutions are likely to fail when addressing the complexity of the issues. This is clear from the present commentary.

Author Response

Thank you for your review of the paper. The authors worked to ensure the review was balanced and represented the need for greater inclusion of all sectors to advance the topic of sustainable diets for all.

Edits and recommendations provided by the reviewers have been made and the manuscript updated.

Sincerely,

Kevin Miller

Reviewer 2 Report

Thank you for the opportunity to read this study. The issue of nutrition is important today. Various promotional campaigns supported by relevant institutions have a great influence on what we eat. Lobbying by food-producing corporations, if applicable.

At the very beginning, I would suggest changing the keywords. They are unsuitable for the article presented.

I understand that the article is not based on proprietary research, but on information obtained from secondary sources. However, it should be written what the purpose of the authors was when they wrote the article.

What was the method of the sources used, according to which key the authors used the given literature items? Were these the latest (most recent) publications on the subject in question?

For a review article, there is very little literature to which the authors refer - only 37 items. An article that does not have a research part should be supported by at least 60 references.

The limitation subsection is missing from the Conclusion section.

Lines starting with 475 should also be removed - these are the remains of the journal form.

Author Response

Hello:

Thank you for the thoughtful review.  We have made edits to reflect the comments from all reviewers.

The objective of the authors was to highlight the need for all sectors to be engaged in finding accessible, affordable, nutrient-dense solutions to sustainable diets. Publications – especially on consumer insights of sustainable diets – are relatively rare in this new, unexplored field.  As a result, there is little research available to add to the topic.  Inclusion of more citations would have unintentionally broadened the scope of the paper.

Yes, we agree manuscripts often include more citations, but the value of a paper should not be determined by abundance of citations - especially when the paper reflects a review and opinion.  Th objective of the manuscript is a call to action for greater inclusion of the food industry to move towards increasingly sustainable and nutrient-dense products and is not presented as a systematic review.

Manuscripts often include “limitations” when there are methodological or interpretive shortcomings possible and are more appropriate for papers with study design and interpretations of results.

Sincerely,

Kevin Miller

Reviewer 3 Report

The concept of sustainability beyond local and fresh foods is important to convey. However, the case for packaged foods as part of a sustainable food system is important to all socio economic levels is important to convey especially as a solution to food waste, access and availability. A broader decision of coalitions including the agricultural industry would strengthen this commentary,  See specific comments in the attached document. 

Author Response

Thank you for the positive comments regarding the subject matter of the paper. Yes, the authors would have liked to invite more participants to the paper, but it is challenging to keep a topic such as this focused when other sectors are included.  As a result, we included recognized subject matter experts on nutrition, sustainable agriculture, food processing, and regulatory to give input.  The intention is to create another paper in the series that can tackle other aspects of sustainable diets, including packaging and affordability. There will be an opportunity to get diverse input from other sectors.

Sincerely,

Kevin Miller

Round 2

Reviewer 2 Report

Dear Autors,

As for Communication, which has the right to not include methodological part, results and discussion with strengths and limitations, it is done reasonably well. I could discuss the availability of literature, but it is pointless to mention that nowadays, in the age of the Global Internet, there is access to many databases, not only in English. I hope the authors will  use them in the future.